# 1 Measurement Report: Real-Time Remote Sensing of the Coastal

## 2 Boundary Layer and its Interaction with Meteorology at Cape Grim,

## 3 Australia

Zhenyi Chen<sup>1\*</sup>, Robyn Schofield<sup>2</sup>, Melita Keywood<sup>3</sup>, Sam Cleland<sup>4</sup>, Alastair G. Williams<sup>5</sup>, Alan

Griffiths<sup>5</sup>, Stephen Wilson<sup>6</sup>, Peter Rayner<sup>2</sup>, and Xiaowen Shu<sup>7</sup>

<sup>1</sup> School of Ecology and Environment, Beijing Technology and Business University, 100048, Beijing, China

<sup>2</sup> School of Earth Sciences, University of Melbourne, 3010, Melbourne, VIC, Australia

<sup>3</sup> Climate Science Centre, Oceans and Atmosphere, CSIRO, 3195, Aspendale, VIC, Australia

<sup>9</sup> <sup>4</sup> Observing Systems and Operations, Bureau of Meteorology, 7330, Smithton, Tasmania, Australia

<sup>5</sup> Australian Nuclear Science and Technology Organisation, Lucas Heights, New South Wales, Australia

<sup>6</sup> School of Earth, Atmosphere and Life Sciences, University of Wollongong, 2522, NSW, Australia

<sup>7</sup>Anhui Institute of Optics and Fine Mechanics, Chinese Academy of Sciences, 230031, Hefei, China

\* Correspondence to: Zhenyi Chen (zychen@btbu.edu.cn)

Abstract. Despite considerable efforts during the last decade, real-time characterization of the marine boundary layer and 16 aerosol optical properties over the Southern Ocean remains scarce. We conducted simultaneous measurements of the marine 17 boundary layer utilizing a synergy of remote sensing technology at the Baseline Air Pollution Station at Cape Grim in 18 northwestern Tasmania, Australia, from 14 May to 16 July 2019. Aerosol optical properties were monitored by lidar 19 (miniMPL) and a ceilometer to identify the boundary layer height, and sodar provided wind profiles to investigate their 20 influences on the layer evolution. Boundary layer heights simulated using the Weather Research and Forecasting (WRF) 21 model were also employed for comparison purposes. Through complementary analyses of three cases representing different 22 source influences (marine, sea breeze and continental), this paper evaluates two algorithms (Image Edge Detection 23 Algorithm (IEDA) and gradient method) for boundary layer height detection and examines the vertical aerosol distribution 24 within the boundary layer at Cape Grim with an emphasis on the contributions of regional and local meteorology. We found 25 IEDA generally performed better than the gradient method, especially during the marine-flow influenced period with a 26 convective layer structure. Different features of boundary layer structures in three episodes, including differential boundary 27 layer growth and interaction with wind evolutionary processes were investigated. One was characterized by a diurnal 28 variation with a boundary layer height of approximately 0.2 - 0.5 km, associated with the veering of the wind vector within 29 the marine boundary layer during the development of a sea breeze. The other showed a thermally stable layer below 0.3 km 30 with an enhanced extinction coefficient and linear depolarization ratio under the influence of continental sources, which was 31 also validated by the observation from Cloud-Aerosol Lidar and Infrared Pathfinder Satellite Observation (CALIPSO) 32 satellite. The increasing extinction coefficient and depolarization ratio with wind speeds may be attributed to the increased 33 wet sea salt production and regional transportation from mainland Australia.

- 34
- 35

36

### 38

#### 39 1 Introduction

The planetary boundary layer (PBL), known as the lowest part of the troposphere, is strongly influenced by the underlying 41 surface and mediates the critical interactions between the atmosphere and the surface. It undergoes large spatial-temporal 42 variability in response to the combined action of mechanical and thermal forcing, in the order of 1-h timescale (Stull, 1998). 43 Many important processes occur in the boundary layer, such as convection, turbulence, and decoupling, making it a vital part 44 of our atmosphere to study and understand. Over the ocean, the marine boundary layer (MBL) has direct contact with the 45 ocean instead of the land and the MBL height determines the extent of marine aerosol vertical mixing, transportation and the 46 resulting concentrations. Therefore, the representation of MBL evolution processes, together with the optical properties of 47 marine particles, has been recognized as an essential ingredient of atmospheric and atmosphere-ocean coupled models. At 48 the Cape Grim Baseline Air Pollution Station (CGBAPS) in Tasmania, the Southern Ocean MBL represents the mixing 49 volume for long-term hemispheric baseline monitoring of greenhouse gases and ozone-depleting substances. 50 The Southern Ocean is a particularly interesting region for MBL observations. Higher than global average wind speeds are 51 sustained over a large and uninterrupted uniform fetch of ocean surface, enabling investigations into the exchange, transport 52 and mixing relationships between MBL and meteorological quantities, as well as various airborne species of interest (e.g., 53 aerosols, trace gases), that are distinct from any other oceanic region (Yuan 2004; Hande et al., 2012). Based on the 54 measurements in the Southern coastal areas, a general understanding of aerosol properties was formulated, based on 55 aerosol/atmospheric dynamics to explain particle formation, evolution, and transport (i.e., Monks et al., 1996; Murphy et al., 56 1998; Bates et al., 2000; Gras et al., 2015; Mace et al., 2018; McCluskey et al., 2018; Bronselaer et al., 2020). For aerosols, 57 large aerosol backscatter, low depolarization ratio, and high relative humidity within the well-mixed near-surface layer were

- obtained at all latitudes across the Southern Ocean (Alexander et al., 2019). The enhancements of cloud condensation nuclei
   (CCN) concentrations were also detected under continental-influenced air masses (Alroe et al. 2020; Simmons et al., 2021).
- Synoptic conditions strongly influence marine aerosols. Particularly the wind speed and wind direction were critical for the
- determination of fresh sea salt concentrations over the coastal city of Sydney, Australia (Crawford et al., 2016). The seasonal
- radon cycles in the mid-Southern Ocean site were also dominated by the synoptic transport of continental air (Chambers et
- al., 2018). However, the limited understanding of the vertical characteristics of marine aerosols and their interactions with
- the MBL remains a significant source of uncertainty for climate simulations.

Many studies have shown that ground-based Light Detection and Ranging (lidar) is effective at deriving global PBL/MBL height distributions over land and ocean, as it is the most straightforward and least expensive technique among remote 66 67 sensing methods (Emeis et al., 2008; Groß et al., 2011; Bravo-Aranda et al., 2017; Su et al., 2017; Tangborn et al., 2021). 68 This optical technique has the least interference with its environment, with the high temporal and vertical resolution, 69 measuring the aerosols and trace gases of the atmosphere. Thus, the PBL/MBL evolution could be inferred by taking 70 aerosols as a tracer from attenuated lidar back-scattered signals. Though lidar measurements have been performed in rural 71 and marine campaigns for PBL/MBL detection (e.g., Hennemuth et al., 2006; Rosen et al., 2000; Peña et al., 2013; Zhou et 72 al., 2015; Luo et al., 2016; Melecio-Vazquez et al., 2018), this is still a challenging task for MBL detection as offshore 73 observations are infrastructurally demanding and difficult to obtain. Studies on the vertical characteristic of marine aerosols 74 and their interaction with the MBL are still scarce.

- This study used systematic, real-time in situ and remote sensing measurements to characterize the MBL process and aerosol
- vertical optical properties at Cape Grim, Australia. On the one hand, we evaluated different methodologies for reliable

estimations of boundary layer height (BLH) from lidar (miniMPL), ceilometer and sodar against Weather Research and Forecasting (WRF) model simulations. On the other hand, the MBL processes and the influence of regional and local meteorology are investigated. The aerosol extinction coefficient and linear depolarization ratio at different heights are

80 discussed, and the interactions between MBL and aerosols are clarified in three case study periods.

#### 81 2 Experimental methods

#### 82 2.1. Sampling Sites and Measurements

The observations were performed from 14 May to 16 July 2019 at the Cape Grim Baseline Air Pollution Station (40.68°S 84 and 144.69°E), near the northwestern tip of the island state of Tasmania, Australia, which is located far from sources of 85 anthropogenic aerosols. A topographic map of the experimental site and the surrounding area is shown in Figure 1a. When 86 the prevailing wind blows from the southwest to northwest sector (190° to 280°), the airflow that influences the station is 87 identified as the baseline sector and typically travels across several thousand kilometers of the Southern Ocean (Lawson et 88 al., 2015). In addition, we use a minimum wind speed of 7 km/h (1.94 m/s) to determine baseline conditions, which is 89 consistent with previous observation at Cape Grim (Cravigan et al., 2015). In northerly wind directions, continental air from 90 Melbourne city some 250 km away is transported across the ocean (Bass Strait) to the station. The near-surface observations 91 of 10-m meteorological variables, wind speed and direction were measured using 4 cup anemometers (Model: 014A, Met 92 One Instruments), and the temperature (T) was obtained using a standard meteo probe (Model: HC2-S3, ROTRONIC). The 93 data collected as minute averages were processed to calculate hourly means. All times are recorded in local standard time 94 (UTC+10). The distribution of the wind speed and direction during the observation is illustrated in Figure 1b. It shows that 95 the dominant wind directions were west and southwest with a probability of 19.8 % and 14.9 %, respectively. Further, we 96 find that the wind speed was more than 8 m/s in about 56.7 % of the period and rarely exceeded 20 m/s.

The measurements included back-scattered signals from a portable eye-safe elastic back-scatter lidar system (MiniMPL, 98 Sigma Space Corporation, USA), a ceilometer (CL51, Vaisala, Finland), and wind profiles from a sodar (Sodar PCS.2000-24, 99 Metek, Germany). The miniMPL measures back-scattered profiles and linear total depolarization ratio (DPR) of atmospheric 100 particles at a 532 nm wavelength with a spatial and temporal resolution of 30 m and 30 s, respectively. The ceilometer 101 operates at a wavelength of  $905 \pm 10$  nm, which makes it sensitive to water vapor in addition to aerosols (Wiegner and 102 Gasteiger, 2015). The receiver records the returned signal, with a temporal resolution of 36 seconds every 10 m. Although 103 the full optical overlap for CL51 is reached at lower ranges than the miniMPL due to its coaxial beam design, others have 104 found CL51 profiles only trustworthy above 70 m (Martucci et al., 2010). Therefore, the attenuated lidar back-scattered 105 signal for miniMPL and ceilometer in our study was calculated above 100 m. The sodar component emits beeps from three 106 antennas at an acoustic frequency of 1290 Hz, providing continuous measurements of wind horizontal and vertical vectors 107 with a vertical resolution of 10 m at10-min intervals. The measurement height ranges from 20 m to 300 m. An extensive 108 description of the miniMPL, ceilometer and sodar can be found in Chen et al. (2019) and Emeis et al. (2018).

#### 109 2.2. Data analysis

#### 110 2.2.1 Boundary Layer Height (BLH) retrieval by miniMPL and ceilometer

Multiple approaches have been developed to identify the BLH from lidar, such as the gradient detection algorithm, curve

fitting (CF), wavelet covariance technique (WCT), and image processing (Flamant et al., 1997; Steyn et al., 1999; Wood and

Bretherton, 2004; Barrs et al., 2008; Welton et al., 2000; Yang et al., 2013). Among them, the gradient method is one of the

most widely used as it does not need a complicated selection of specific parameters as required by CF or WCT. In most cases,

the first significant peak of the gradient profile is regarded as the upper limit when searching for the boundary layer top, and 116 the deepest valley below it is then identified as the BLH (Lewis et al., 2013). On the other hand, image processing based on 117 overall image detection can provide a more continuous BLH and is not easily affected by clouds or the residual layer 118 (Vivone et al., 2021; Xiang et al., 2019). Our study utilizes the Image Edge Detection Algorithm (IEDA) based on 119 mathematical morphology. The main steps of PBL height retrieval by IEDA are illustrated in Figure 2. Specifically, the 120 original back-scattered signal was first obtained from the lidar (Figure 2a) and converted to the 256-order grayscale color 121 image (Figure 2b). Secondly, it was convoluted with the two-dimensional Gaussian kernel, and the result was taken as the 122 output pixel value (Figure 2c). According to the difference in the grayscale feature between the target and the background 123 region, a threshold value, recommended by Morille et al. (2007), was automatically defined to separate them. Thirdly the 124 mathematical morphology-based image edge processing was implemented, with the corrosion and expansion, to extract the 125 corresponding morphology of the image. The most considerable change in the edge points was then obtained and connected 126 to make the edge of the image (Figure 2d). Finally, the actual value of the BLH was calculated from the position of the target 127 edge of the image after eliminating isolated, non-connected edges (Figure 2e). More details could be found in Xiang et al. 128 (2019).

Figure 3 illustrates the BLH retrieval on 17 May using IEDA and gradient method from the ceilometer and miniMPL, 130 respectively. In the data processing, the IEDA gives the best results most of the time; however, it doesn't work when there is 131 a sporadic signal with a low SNR from the ceilometer (see solid black line in Figure 3a). In comparison, the gradient method 132 exhibited more sensitivity with diffuse aerosol layers but could give false results when multiple-layer clouds are present (see 133 green circles in Figure 3b) from miniMPL. Therefore, considering the analysis in an automated way and providing 134 continuous results with good reliability, without losing detailed evolution of the boundary layer, the IEDA and the gradient 135 detection algorithm was chosen as optimum algorithms for the miniMPL and ceilometer, respectively. Regardless of the 136 method we employed, prior to the image processing or determination of gradient minima, background subtraction and pulse, 137 overlap, and range corrections were applied to the raw back-scattered data to derive the normalized signal with arbitrary 138 units. For cloud screening, a selected window was gliding from the ground and the integral of signals in the window was 139 calculated (Platt et al., 1994). After comparing with the "threshold" value, the random noise was eliminated and the cloud 140 information could be distinguished from aerosols. A manual quality assurance step was finally performed to identify whether 141 the BLH was implemented correctly.

#### 142 **2.2.2 WRF model set up**

In this study, the WRF model, version 4.0, was applied to assess the BLH. Daily model simulations were computed with a 144 36-h forecast cycle, with a 12-h spin-up cycle added to counter instability issues with the simulation. The simulation domain 145 was centered at the observing site at Cape Grim, and a two-way interactive nest with three horizontal resolutions of 18, 6 and 146 2 km were used. The parent domain extended over to the Bass Strait and includes parts of the Tasmania and the Indian Ocean 147 (latitude:  $38.5^{\circ}S - 42.5^{\circ}S$ , longitude:  $142^{\circ}E - 147^{\circ}E$ ). A 2 km spatial resolution domain was defined for the nested domain 148 that covers the Cape Grim region (latitude: 40°S - 41.2°S, longitude: 143°E - 145°E). This innermost nested grids were 149 utilized to compare BLH simulations at the nearest grid to our Cape Grim observing site. The main physical options used 150 here include rapid radiative transfer model (Iacono et al., 2008), and the Noah land surface model (Tewari et al., 2004). 151 Previous studies (Bossioli et al., 2009; Banks et al., 2015) have evaluated the WRF model-simulated BLH using different 152 PBL schemes and indicated that the most accurate simulations of BLH are from a non-local scheme. In order to obtain an 153 efficient set up of the WRF model configuration in terms of the lowest error in BLH, three PBL schemes including two 154 nonlocal closure schemes (YSU and ACM2) and one local closure schemes (MYJ) accompanied by their relevant

- surface-layer schemes were tested. The results were compared against the BLH from the miniMPL, showing that though all
- model runs estimated the BLH with a systematic negative or positive bias, the best correlation with observed BLH was given
- to YSU scheme (not shown here). Therefore, only the WRF simulation with YSU scheme was applied in this paper. The
- initial and lateral boundary conditions for meteorological simulation were generated from the National Center for
- Environmental Prediction (NCEP) Final Operational Global Analysis data with a 1°×1° spatial resolution and were updated
- every 6 h. The simulation was run with 35 vertical layers extending up to 50 hPa. The final output was provided every hour.
- Further details are summarized in Table 1.

#### 162 2.2.3 CALIPSO satellite

In order to validate the aerosol type from the miniMPL retrieval, the vertical distribution of DPR was also analyzed from the 164 Cloud-Aerosol Lidar with Orthogonal Polarization (CALIOP) instrument on board the Cloud-Aerosol Lidar and Infrared 165 Pathfinder Satellite Observation (CALIPSO) satellite. CALIOP aerosol profile products (e.g., Level 2 and 3) have been 166 widely used to provide new insight into the role that atmospheric aerosols and clouds play in the Earth's climate change 167 process (Misra et al., 2012; Mehta and Singh, 2018; Kulkarni and Sreekanth, 2020). Among the data products, the Level 2 168 data provides profiles of total attenuated backscatter coefficients at wavelengths of 532 and 1064 nm, and two perpendicular 169 and parallel polarization components at 532 nm. In the present study, the DPR at 532 nm from CALIOP level 2 data (version 170 4.20), has a horizontal resolution of 5 km and the vertical resolution of 60 m from the surface to 20 km (Kim et al., 2018), 171 were utilized to validate the aerosol type. During the case study period in section 3.2, two overpasses on 21 June and 26 June 172 are studied. The DPR profiles were averaged over  $\pm 0.1^{\circ}$  latitude and  $\pm 0.1^{\circ}$  longitude closest to Cape Grim. The horizontal 173 distances between the satellite footprint and our miniMPL site were less than 100 km.

#### 174 3 Results and discussion

#### 175 3.1 An overview of winter BLHs over Cape Grim

By utilizing the algorithm described above, we calculated the box chart of winter-time BLHs from the miniMPL (IEDA) and 177 ceilometer (gradient) measurements over Cape Grim from 14 May to 16 July 2019 in Figure 4a. Only days with these two 178 instruments operating simultaneously were considered. Therefore, a total of 43 days of data were calculated. Generally, the 179 BLH at Cape Grim hardly exceeded 1 km height in this period. We can see that the BLH for miniMPL (IEDA) was featured 180 as the normal distribution, with the mean/median value of 0.48 km. In comparison, the results from ceilometer (gradient 181 method) show more variability, with higher averaged BLH values of 0.63 km due to multiple sharp gradients corresponding 182 to multi-layer or lofted aerosol layers or clouds. Similarly, Figure 4b illustrates that the diurnal variation of BLH from the 183 miniMPL revealed a relatively smooth cycle, while those from the ceilometer varied more and were generally higher. The 184 maximum BLH was mostly observed near noon for both instruments. For the miniMPL, the maximum BLH (0.52 ± standard 185 deviation of 0.087 km) was observed with a high BLH persisting for the entire day. In contrast, ceilometer-derived BLH 186 reached a maximum value,  $0.74 \pm 0.043$  km, at 11:00 LST and appeared to collapse immediately afterward before sudden 187 significant growth at 22:00 LST. The standard deviation of BLH diurnal variation ranged between 0.012 and 0.142 km and 188 0.039 and 0.184 km from the miniMPL and ceilometer, respectively. Good agreement occurred between 15:00 and 19:00 189 LST was likely related to lower turbulence (reduced intermittent aerosol gradients) in the afternoon.

#### 190 3.2 Selected MBL evolution and aerosol observations: Case study period

- The period from 20 June to 27 June 2019 was selected for the case study (Figure 5) because it consisted of the onset of a sea
- breeze episode (E1), sea breeze and offshore flow interaction period (E2) and offshore flow episode (E3) based on synoptic

analyses (surface, 850 hPa, Figure 6) in combination with typically measured wind conditions from sodar, near-surface 194 wind/air temperature from the collocated automatic weather station (Figure 7) and different back trajectories (Figure 8). Due 195 to the irregular shoreline of the measurement area, flow from the southwestern to northwestern directions (190° to 280°) is 196 indicative of onshore flow, while flows from other directions are indicative of offshore flow. The sea breeze occurrences 197 were identified using the criteria based on wind direction, wind speed and dew point temperature (Caicedo et al., 2021). 198 During sea breeze episodes E1 and E2, wind speed varied considerably (ranging from 4 to 12 m/s, see Figure 5b) most of the 199 time with a moderate temperature (less than 10 °C, see Figure 5c). In contrast, offshore episode E3 was characterized by 200 increasing temperatures (as high as 14 °C), high wind speeds (ranging from 7 m/s to 14 m/s) and predominantly northeastern 201 winds, excluding an occasional twenty-minute northern wind in the morning. Figure 5d shows the averaged 10-minute BLHs 202 from miniMPL(IEDA) and ceilometer (gradient). Both the miniMPL and ceilometer presents the moderate values of the 203 BLH depth (of the order of 0.5 km) from 20 June to 23 June while the miniMPL reveals more diurnal characteristic of BLH 204 with the veering of the wind vector (from southwest to more westerly marine flow) observed by the measurements close to 205 the surface (See Figure 5a). From the noon of 23 June, the BLH experienced a gradual decrease when the wind was 206 characterized by the transition from sea breeze (west-southwestern) to land flow (northerly). Then the agreements between miniMPL and ceilometer BLH observations improved when northerly winds prevailed under offshore flow from 24 June to 207 208 27 June. It is noticeable that from 20 June to 24 June, the ceilometer yielded higher BLH results than the corresponding ones 209 from miniMPL because when the convective boundary layer began to form near the surface during this marine-affected 210 period, several residual aerosol layers usually existed aloft. These aerosol layers introduced large signal gradients in the 211 detection algorithm, and lead to over-estimations of the BLH for the ceilometer.

#### 212 3.2.1 Synoptic conditions

The synoptic pattern on 20 June 2019 during the sea breeze period consisted of a cyclone (low-pressure) system over the 214 northern Tasmanian region and its associated anticyclone (high-pressure) system centered over the Indian Ocean moving east 215 (Figure 6a). The anticyclone system was sustained mainly to the west of our site for the next two days and crossed over on 216 23 June (Figure 6b). Therefore, the pressure increased from 1029 to 1034 hPa with the approaching anticyclone center. The 217 typical rising temperatures, low cloud cover, and weak winds, within the center of an anticyclones system, usually facilitate 218 the formation of sea breezes. On 27 June, the area experienced a decrease in pressure and the cold air outbreak from the west. 219 The weather stabilized subsequently (Figure 6c). On that day, no sea breeze developed because of the strong northeastern 220 wind.

#### 221 3.2.2 Backward trajectories

We analyzed the characteristics of aerosols related to the classified back trajectory to estimate the source of aerosols preliminarily. The HYSPLIT (HYbrid Single-Particle Lagrangian Trajectory) model (Draxler and Rolph, 2003) was configured to examine the various air mass pathways and aerosol sources. Each member of the trajectory ensemble was calculated at heights of 0.5 km above sea level. After the simulation of the 72-hour trajectories, we found that air masses originated from the Southern Ocean and entered the measurement site along the west coast of the Cape Grim in pperiod E1 (Figure 8a) and E2 (Figure 8b). Period E3 (Figure 8c) presented air masses traversing mainland Australia before they arrived at our observing site.

#### 229 **3.2.3 BLH evolution under different sources**

- We chose 20 (E1), 23 (E2) and 27 (E3) June 2019 as the specific days ("episodes") during the selected case study period to
- further investigate the MBL evolution and aerosol properties. Figure 9a shows the time-height aerosol back-scattered signal

intensity obtained on 20 June 2019 by the miniMPL. The dominant characteristic of this day was the strong diurnal pattern of 233 MBL development. BLH ranged between 0.2 km and 0.5 km, with thermally stable conditions (low BLH) during the 234 morning and evening, and unstable conditions (high BLH) during the day. The MBL evolution was quasi-stationary in the 235 early morning period (02:00 - 08:00 LST). Then it became more pronounced in the convective period (09:00 - 14:00 LST) 236 with an increasing hourly growth rate (from ~ 30 to 110 mh<sup>-1</sup>) and reached a maximum height, 0.56 km, at 14:50 LST. After 237 18:00 LST, the BLH decreased again due to the substantial reduction in convective turbulence, with a new nocturnal 238 boundary layer developing at an altitude of approximately 0.3 km. From Figure 9b and 9c, the sodar captured the pronounced 239 increase in vertical wind components with height in the morning. The most drastic speed increase occurred at 09:00 LST, 240 indicating the onset of MBL development. The southeasterly winds from the land dominated the entire area from the 241 morning hours, while during the afternoon hours, the intrusion of a flow of southwesterly sea breeze disrupted the easterly 242 winds.

From the surface meteorological parameters in Figure 7a, increases in the temperature and wind speed after sunrise were also observed, while the wind direction changed from southeast to southwest at 09:00 LST, which coincided with the sodar results. The WRF estimates were lower than those from the lidar in the early morning (before 06:00 LST). They matched the BLH estimates from the lidar until 10:00 LST. It seems the initial rising trend of BLH is captured well by WRF. However, the model results showed an increasing BLH at 11:00 LST and maintained a high depth in the afternoon and nighttime. This overestimation of BLH WRF may be attributed to the non-accurate prescription of surface roughness and induced turbulence intensity in the atmospheric surface layer.

On 23 June 2019, a weak breeze period interacted with the uplifted strong local offshore wind with convectively active MBL 251 heights ranging from 0.25 to 0.45 km (miniMPL) and 0.18 to 0.82 km (ceilometer) (Figure 10a). Compared with E1, the 252 wind condition in this case presented a similar trend with a more significant difference in winds between the ground and 253 higher heights (Figure 10b and Figure 10c). In the morning, the horizontal winds were calm near the surface and mixed with 254 a robust offshore flow (southeastern wind) above up to 0.2 km until 13:00 LST. Then, the winds returned to the onshore 255 direction with reducing wind speeds near the surface (seen in Figure 7b), while the winds from 0.2 to 0.3 km were relatively 256 strong (3-6 m/s, Figure 10b). Wind flows transitioned back to offshore again by ~ 22:30 LST. Comparing the E1 and E2, 257 wind observations indicated features of boundary layer jets characterized by strong vertical shear, weak horizontal shear, and 258 noticeable diurnal variation. More specifically, the increased wind speeds (> 10 m/s at a 0.3 km height in both episodes) 259 associated with a strong baroclinic low-level jet dominated the area from 20 to 23 June. High turning of the wind and 260 relatively high wind speeds were observed at heights far above 0.1 km. The wind at these heights was marine, and the 261 increase in turning was related with the increasing wind speed. Meanwhile, the local offshore winds (southeastern) interacted 262 with the developing southwestern sea breeze, resulting in the wind vector diurnal veering in the morning, and the sea breeze 263 returned to the area in the afternoon. Accordingly, the MBL usually developed within the complicated interaction period, and 264 the BLH was determined by the depth of the inflow. A continuously 0.2-0.5 km thick layer with marine particles was present 265 from the onset of the sea breeze period on 20 June and remained constant during the mature sea breeze period on 23 June, 266 pushing marine aerosols upward. The WRF-derived BLH calculations showed similar behavior as E1; the BLH was low in 267 the early morning, increased during the daylight period (10:00 -16:00) and finally declined and stabilized during the night. 268 However, E2 presented a shorter period of MBL depth at noon and then rapidly decreased when the boundary layer jet 269 occurred.

The lidar back-scattered intensity on 27 June with pure offshore winds showed significant scattering close to the surface,

implying a layer of high aerosol concentration was present at lower ranges between 0.23 km and 0.38 km throughout the day

(Figure 11a). The much higher extinction derived from the lidar throughout the day (highest extinction value of 0.1 km<sup>-1</sup>) 273 was correlated with higher aerosol concentrations from continental sources, while the sea breeze flow (in E1 and E2) 274 transported low aerosol concentration air. No sea breeze characterized that day due to the northernly (~330°) wind. The sodar recorded the strong northwestern wind from 0:00 to 03:00 LST and the prevailing northeastern winds afterward 275 276 (Figure 11b and Figure 11c). The surface wind speed revealed large variations from 7 m/s to nearly 14 m/s and the temperatures were higher with values from 10 to 13 °C (Figure 7c). Under the influence of constant offshore flow, the 277 278 turbulence produced from surface heating and cloud-top radiative cooling became sufficient to maintain a more 279 homogeneous and mechanically well-mixed MBL. The WRF-based results compared better to the observations than the 280 behavior in the sea breeze examples. The comparison was better in the daytime 06:00-18:00. After 18:00 onward, the WRF 281 MBL height continued to be higher than the observed values.

#### 282 3.2.4 Interactions between the MBL and vertical distribution of aerosols

Figure 12 presents comparison of the 532 nm DPR (red line) and extinction coefficient (blue line) from ground-based miniMPL against nearly co-located CALIPSO DPR observations (green line). For miniMPL, thirty-minute mean profiles of the aerosol extinction coefficient and DPR from 12:45 to 13:15 LST on 20 (E1), 23, and 27 June 2019 were calculated. For CALIOP, DPR profiles stand for the time-averaged from 04:30:07 to 04:52:38 on 21 June and from 04:30:07 to 04:52:40 on 26 June 2019. According the previous measurements (Müller et al., 2007), constant lidar ratios, 23 sr and 60 sr (at 532 nm), were chosen for marine and continental aerosols, respectively. Error bars indicate the lidar retrieval uncertainty.

The DPR, defined as the ratio of the received lidar signals in cross and parallel channels, indicates whether the particles have 290 a nonspherical shape (Tesche et al., 2009). Understanding the variation in DPR is essential to help distinguish aerosol types 291 roughly and identify potential sources. Under maritime conditions, the relative contribution of coarse particles, though 292 variable, is generally higher than those over urban or industrial areas but lower than desert dust (Dubovik et al., 2002). 293 However, in our case, the DPR values from miniMPL during sea breeze events were relatively low, ranging from 0.01 to 294 0.02 (E1) and 0.02 to 0.03 (E2) below 1 km. They remained constant with height, suggesting that the majority of particles 295 were typical spherical sea salt or wet sea salt (above approximately 65-70% RH) suspended in the atmosphere. The enhanced 296 extinction coefficient with a slightly increased value of DPR in E2 was related more to the interaction of sea breezes and 297 strong local flow conditions with oceanic and other types of marine aerosols, i.e., some dry marine aerosols. In comparison, a 298 much larger depolarization ratio (ranging from 0.2 to 0.3) accompanying an increased extinction coefficient (up to 0.3 km<sup>-1</sup>) 299 was found under continental source influences in E3. For the profile-to-profile comparison, the aerosol DPR feature obtained 300 by CALIOP was similar to miniMPL except for the aerosol layer above 1.5 km as the aerosol backscatter from miniMPL 301 observation over this altitude was not strong. During the marine-flow-influenced E1 period (but 21 June for CALIOP, green 302 line in Figure 12a), aerosol DPR ranged between 0.02 to 0.05 within the MBL, larger and noisier than the corresponding 303 profiles from miniMPL due to the cloud contamination. On 26 June, in the lower troposphere, the DPR values varied from 304 0.14 to 0.38 from the surface to 1.5 km, with better coincidence with corresponding profiles from miniMPL.

#### 305 4 Conclusion

In this research, we evaluated two algorithms for BLH estimations based on miniMPL (IEDA) and ceilometer (gradient method) observations against WRF model in winter Cape Grim, 2019. For algorithm comparison, the IEDA is able to distinguish between the different layers, while the gradient method is not so feasible when the elevated/aloft aerosol layer existed. It is reasonable that IEDA outperforms the gradient method considering that IEDA is based on the overall image and not easily affected by clouds or the residual layer. The reliability of the gradient method from ceilometer largely depends on

- the different atmospheric conditions. The first condition corresponds to atmospheric aerosol from local sources and a convective MBL with stratification, when the ceilometer usually overestimated the miniMPL. The second scenario is related to aerosols from a non-local source decoupled from the MBL with an underlying well-mixed PBL. Under such condition, BLH from miniMPL was in better agreement (difference lower than 50 m) with the ones obtained from the ceilometer. Similarly, for WRF simulation comparison, the largest differences between model and observations were also found during the sea-breeze synoptic flows with more dynamic MBL conditions. In general, except in the early morning, the simulated BLHs show WRF tended to have a positive bias with higher values of BLH than the observations. It indicates that the WRF
- BLH simulation for the coast region still remains a challenge.

For the sea breeze and local flow interaction period, it was reasonable that wet sea salt particle production above the humid 320 ocean resulted in a slight increase in the DPR without aerosol crystallization (RH > 75%). For the continental 321 source-influenced period, the significant increase in DPR was more associated with soil mixed with other nonspherical 322 aerosols after long transportation, indicating that aerosols of continental origin could be present in the Cape Grim MBL. 323 Therefore, our results highlight the importance of mixed particles with high DPR values brought from continental sources 324 from long-transported anthropogenic sources and regional transport to aerosols lofted within the MBL. We also observed 325 dynamic vertical structures of wind vectors below 0.3 km, closely interacting with the MBL evolution. Thus, more 326 intercomparison under different meteorological conditions and varying parameters for the chemical and aerosol observations 327 in the lower MBL are proposed to better understand the MBL and marine aerosols.

#### 329 Data availability

The miniMPL, ceilometer, soda data and surface meteorological data will be openly available in Mendeley Data (doi: 10.17632/gtpp4ydfk6.1., Chen et al., 2022). The source code of the WRF-Chem model is archived on UCAR data repository (http://www2.mmm.ucar.edu/wrf/users/download). The HYSPLIT model can be acquired from the NOAA Air Resources Laboratory (http://www.arl.noaa.gov/ready/hysplit4.html).

#### 335 Author contributions

Zhenyi Chen wrote the manuscript and made the overall data analysis. Robyn Schofield provided miniMPL measurements.
Melita Keywood and Sam Cleland provided the ceilometer and sodar data. Zhenyi Chen and Yan Xiang designed the
automatic algorithm for boundary layer height retrieval. Alastair Williams, Stephen Wilson, Alan Griffiths, and Xiaowen
Shu provided academic support.

#### 341 Competing interests

- The authors declare that they have no conflict of interest.

#### 344 Acknowledgement

This work was supported by Research Project Agreement between University of Melbourne and CSIRO. The author gratefully acknowledges the effective cooperation of colleagues of Melbourne University and CSIRO. Robyn Schofield, Sam Cleland and Melita Keywood provided the raw data and relevant documents. Alan Griffiths helped to improve the IEDA development. The discussion with Alastair G. Williams and Stephen Wilson was very valuable for the data analysis. The staff who assembled, tested and calibrated the miniMPL, ceilometer and sodar systems are also acknowledged. The Cape

Grim Baseline Air Pollution Station is funded and managed by the Australian Bureau of Meteorology, and the AGAGE

scientific program is jointly supervised with CSIRO Oceans and Atmosphere.