# Peer review of "Measurement Report: Real-Time Remote Sensing of the Coastal"

_Atmospheric Chemistry and Physics, 2022_

## Referee Comment (RC1)

**Measurement Report: Real-Time Remote Sensing of the Coastal Boundary Layer and its Interaction with Meteorology at Cape Grim, Australia**

Zhenyi Chen, Robyn Schofield, Melita Keywood, Sam Cleland, Alastair G. Williams, Alan Griffiths5, Stephen Wilson, Peter Rayner, and Xiaowen Shu

This paper presents measurements from a lidar, ceilometer and sodar at Cape Grim, from which the boundary-layer height is derived. Although Cape Grim is an interesting station for measurements of pristine Southern Ocean air, it is not clear to me what the purpose of this paper is meant to be. Most of the results show three case studies which are little more than examples – they don't lead to any useful conclusion beyond this particular dataset.

At the heart of the difficulties encountered in this paper, and not discussed at all in the introduction, is what is meant by the BLH over the ocean and how this relates to the distribution of aerosols. The BLH is essentially a thermodynamic concept, but the marine boundary layer can be stable, inhibiting the vertical transport of aerosol. To effect a meaningful comparison with WRF, for example, one would need to understand what WRF means by BLH and whether that is relevant to the distribution of aerosols (i.e. are they likely to be mixed throughout the BL or confined to low levels by inversions in the temperature profile?). The issue is compounded here by the very different 'BLH' derived from the two remote sensing instruments, strongly suggesting that a simple definition is inappropriate.

Regarding the conclusions of the paper:
1. You shouldn't validate measurements against a model!
2. The comparison of the two algorithms is superficial and the discussion of fig 3 (see below) is misleading. The IEDA gives smoother curves because I suspect it is designed to do so, but there is no evidence that it is 'better' than the gradient method.
3. The effect of clouds on BLH measurements is barely mentioned, other than to state without proof that the IEDA method is better in this regard.
4. The result that DPR for pure oceanic air masses is lower than that for a continental air mass with a dust component is hardly new.

For this paper to be publishable, the authors must decide what scientific story they want to tell and structure their paper accordingly. I'm aware that this is a measurement report, but even so the standard of presentation falls well below what is expected, and the relevance of the measurements to atmospheric science generally (rather than locally to Cape Grim) is not explained.

**Detailed comments (major)**

The authors present in fig.3 a derivation of the BLH using the IEDA and gradient methods for 17 May 2019, concluding that the former method is best for the lidar and the latter for the ceilometer. I find this conclusion difficult to reconcile with the figure. Taking the left hand panel, the BLH is around 0.7 km from 3 to 12 h, falls to around 0.4 km by 15 h where it remains for the next few hours, before rising to a maximum around 0.5 km at 21 h and falling sharply to around 200 m at 24 h. The IEDA data, where shown, are consistent with the circles on the left hand panel. Contrast this with the right hand panel, where the IEDA line from 6 – 12 h is around 0.8 – 0.9 km, gently falling to around 0.7 km by the evening. This is almost twice the height from the ceilometer!! Furthermore, many of the circles on the right hand plot (notably between 14 and 18 h) are consistent with those on the left hand plot. The authors do not even comment on this discrepancy, and based on this evidence I conclude that they do not have a reliable method of deriving BLH from their data.

According to the final sentence of p.5, the shaded period E1 in fig 5 contains a sea breeze, and E2 a 'sea breeze and offshore interaction pattern', whatever that is. It is not clear at all how the authors come to these conclusions – how do they define a sea breeze? A reference to Caicedo et al 2021 is given but surely the criteria are not that complex? Then on l.198, we are told that during E1 and E2 the wind speed varied from 4 to 12 m/s, which is not consistent with the diagram. In E1 the wind varies from 2 to 8 m/s and in E2 from 2 to 6. The peak of 12 m/s occurs in the unshaded section between them, during westerly flow. During E3 the maximum wind speed in 5b is 12m/s, not 14 m/s as on l.200. On l.203 the wind vector is said to veer from SW to W from 20 to 23 June, but that isn't what is shown on 5a. Finally, the statement on l.208 that the ceilometer BLH were higher than the lidar BLH from 20-24 June is only true some of the time – at other times the ceilometer BLH is lower. Quite simply, this whole paragraph is inconsistent with the data presented and reflects carelessness on the part of the authors. Furthermore, it only becomes evident in section 3.2.3 why the periods E1, E2 and E3 were selected – this should be mentioned at the beginning of 3.2.

The interpretation in section 3.2.3 leaves much to be desired. For a start, it is repetitive and doesn't obviously make a point. E1 and E2 were periods of sea-breeze flow, so it is not surprising that the signal of land convection is seen as the air sloshes out to sea and back. On l. 244 the wind direction is said to change from SE to SW at 0900 on the 20th, which is simply not true – the change in direction is around 50°, not 90°, and the sodar shows a change to southerly (U=0). On l.247 a throwaway sentence attributes the overestimate of BLH in WRF to 'non-accurate prescription of surface roughness and induced turbulence', with no evidence at all. How did the 'weak sea breeze interact with the uplifted strong offshore wind' (l.250)? What does 'interact' mean here? On l.272 reference is made to lidar extinction – how was this measured? We've only seen backscatter mentioned up to now.

This is further compounded in 3.2.4 by a complete section on extinction coefficient. What algorithm was used to derive it? The backscatter signal does not give the aerosol backscatter coefficient (and hence the extinction coefficient by the assumed lidar ratio) directly. BLH may be derived from the signal profiles, but extinction coefficient requires a retrieval (as indeed is mentioned on l.288).

**Detailed comments (minor)**

p.4 l.120 and fig.2. More details are required of the IEDA algorithm as it is not possible to reproduce this study from the description given. For a start, axes and axis labels are needed in Fig 2 so that we can see the scales being discussed. Secondly, the colour scales should be shown, especially the grayscale in panel b where the white colour seems to correspond to an intermediate value of backscatter (why?). Thirdly, explain better how you go from 2b to 2c – a Gaussian kernel smooths a plot and you take a difference between this and the original plot in some way but I'm not sure how. I realise you give a reference to Xiang et al, but this paragraph needs to be clearer. And do you really mean 'corrosion' on line 124?

p.4 l.133 I can't see any green circles in fig 3b, though I can see a lot of scatter in the gradient method

p.6 l.222. You haven't analysed the characteristics of aerosols, you have used a trajectory model to calculate the source region of the air masses that passed over Cape Grim.

p.6 l.226 period

p.7 l.238 the nocturnal BLH is more like 200 m

p.7 l.239 what is special about 0900? The change of wind with height is no different to the preceding 9 hours; if anything it is less

p.7 l.259 why is the low level jet baroclinic? The front is nowhere near Tasmania.

---

## Author Comment (AC1)

**Response to the Comments from Referee #1**

2

1

We sincerely thank the reviewer for the valuable feedback that we have used to check our newly-developed algorithm and improve the quality of our manuscript as well. The reviewer comments are laid out below in *italicized font* and specific concerns have been numbered. Our response is given in normal font.

7

8 This paper presents measurements from a lidar, ceilometer and sodar at Cape Grim, from 9 which the boundary-layer height is derived. Although Cape Grim is an interesting station for 10 measurements of pristine Southern Ocean air, it is not clear to me what the purpose of this 11 paper is meant to be. Most of the results show three case studies which are little more than 12 examples – they don't lead to any useful conclusion beyond this particular dataset.

13

At the heart of the difficulties encountered in this paper, and not discussed at all in the 14 introduction, is what is meant by the BLH over the ocean and how this relates to the 15 16 distribution of aerosols. The BLH is essentially a thermodynamic concept, but the marine 17 boundary layer can be stable, inhibiting the vertical transport of aerosol. To affect a meaningful comparison with WRF, for example, one would need to understand what WRF 18 means by BLH and whether that is relevant to the distribution of aerosols (i.e. are they likely 19 20 to be mixed throughout the BL or confined to low levels by inversions in the temperature profile?). The issue is compounded here by the very different 'BLH' derived from the two 21 22 remote sensing instruments, strongly suggesting that a simple definition is inappropriate.

23

**24 Main comments**

25 *Regarding the conclusions of the paper:*

26 1. You shouldn't validate measurements against a model!

**Response:** Thank you for the comments. In our revised version, we included the ERA5
reanalysis product, as an additional and straightforward check on our observations and WRF.
Unfortunately, no radiosonde was launched for this campaign. We have added Figure 4, Figure
5 for statistical analysis of the whole observing period and re-plotted Figure 9, 10 and 11(now
presented as Figure 10,11 and 12 in the revised version). Figure 4, Figure 5 and Figure 10-12
are shown as follows.

**34 BLH comparison against ERA5 reanalysis data**

The inter-comparison 1-h averaged BLHs for the miniMPL and ceilometer against ERA5 35 36 during the whole observing period are presented in Figure 4. For miniMPL, an excellent concordance is found between IEDA- and ERA5- derived BLHs, with a correlation coefficient 37 38 of 0.78 (Figure 4a). The gradient method underestimates the BLH with largest negative bias of 39 0.83 km, though its coefficient value is slightly lower (0.71 in Figure 4b). It is probably because 40 the gradient estimates appear to detect the largest negative gradient from the bottom-up. Similar to the miniMPL, the ceilometer generally provides lower BLHs compared to the ERA5, though 41 42 some unidentified elevated aerosol layers result in a few points with much higher BLH than ERA5-BLH. However, the gradient method (Figure 4d) outperforms (Pearson's r = 0.50) the 43 44 IEDA (Figure 4c, Pearson's r = 0.41) against ERA5 BLHs. The discrepancy and uncertainties between the miniMPL/ceilometer and ERA5 can be mainly attributed to (1) the different 45 definitions of the BLHs applied to each method (i.e. edge/gradient detection from aerosol back-46 47 scattered signal for miniMPL/ceilometer, bulk Richardson method for ERA5), (2) the different air masses for the spatial separation of the observing sites, (3) the coarse resolution of ERA5 48 49 and (4) the presence of the lofted layer or cloud layers. Therefore, no single approach can cover 50 all situations over this campaign. Among these causes, (1) (2) and (3) would influence our whole period. More details could be found in the section 3.2.1 of the revised version. 51

53 Figure 4. Comparison of 1-h averaged BLH estimations based on different instruments and

54 methods against ERA5 results: (a) IEDA from miniMPL (b) the gradient method from 55 miniMPL (3) IEDA from the ceilometer and (4) the gradient method from the ceilometer. In 56 each case, a linear regression through the origin is performed (red line) and statistics are shown: 57 slope, Pearson's linear correlation coefficient (Pearson's r), slope, and number of samples (n). 58 The black dashed line is a 1: 1 line.

59

60 Diurnal cycle of BLH

Considering the comparison in Figure 4 above, we chose IEDA for miniMPL and gradient 61 62 method for ceilometer respectively to investigate the diurnal cycles of BLH (Figure 5). Generally, the ERA5 presents smoother and higher averaged BLHs (0.71±0.08 km) than those 63 64 from miniMPL (0.64±0.06 km) and ceilometer (0.65±0.07 km). However, the diurnal cycles of BLH from 1 h-averaged miniMPL (IEDA) observations show good agreement with those from 65 ERA5, especially from the early morning to 11:00 LST, with a negative bias (- 0.02 to -0.10 66 67 km). The miniMPL (IEDA) BLHs reached the maximum of 0.76 km at 14:00 LST while the ERA5 BLHs peaked at 0.86 km at 15:00 LST. In comparison, the ceilometer (gradient) shows 68 69 more variable BLHs, as expected due to multiple sharp gradients corresponding to multi-layer 70 or lofted aerosol layers, and presents less diurnal characteristic of MBL (marine boundary layer). Its BLH fluctuated from 0.52 to 0.72 km in the morning and appeared to collapse immediately 71 again. The largest difference between 72 afterward before growth at 19:00 LST 73 miniMPL/ceilometer and ERA5 occurs during the MBL developing period (from 12:00 to 20:00 LST) and the mean nocturnal boundary layer are higher than 0.5 km. 74 75

---

## Author Comment (AC2)

**Response to the Comments from Referee #2**

We sincerely thank the reviewer for the valuable feedback and the references that we have used to check our newly-developed algorithm and improve the quality of our manuscript as well. The reviewer's comments are laid out below in *italicized font* and specific concerns have been numbered. Our response is given in normal font.

**6 Summary**

Chen et al. use a mini Micro Pulse Lidar and ceilometer in order to document diurnal changes 8 in the boundary layer height at a coastal location in Australia which is subjected to maritime 9 and continental airmasses. They document three case studies, which are representative of 10 distinct air mass sources, in order to understand the observed boundary layer structure. Chen 11 et al. supplement the observations with WRF simulations and discuss some of the similarities 12 and differences between observations and simulations. The authors also employ sodar 13 observations in the lowest few hundred metres in order to quantify small-scale wind changes 14 and their role in altering the boundary layer.

Overall, the manuscript presents new results in a poorly-sampled region of the world, where 16 additional observations and insights of the boundary layer may help with constraining in 17 model simulations some of the well-known issues over the Southern Ocean. Yet, there are some 18 major problems with the manuscript in its present form which need to be addressed 19 comprehensively prior to consideration for acceptance. These revolve mainly around incorrect 20 (and incomplete) algorithms to properly extract the correct boundary layer height (BLH), 21 which is central to the whole manuscript, an ill- defined 'manual checking' procedure, and an 22 apparent lack of cloud screening and lidar / ceilometer backscatter profile removal before 23 BLH detection algorithms are implemented. Idetail my comments below.

The full citations of literature (those not already cited in your manuscript) referred to inmy
comments below can be found at the end of this review.

**27 Major Comments**

1. lines 129 - 135. I disagree strongly with the statements about the performance of the 'IEDA'
and 'gradient method' algorithms for detecting BLH. Indeed the results which you presented in

Figure 3 indicate your algorithms as presently developed and implemented are not satisfactory for detecting the BLH on partially cloudy days. Further, I do not agree with your choices or justifications of using the IEDA for the miniMPL and the gradient method for the ceilometer data. I explain why, and offer suggestions, in the paragraphs below.

Crucially, and I feel this point is glossed over in the manuscript, you cannot compute a BLH 35 via either of your methods (IEDA, gradient) in the presence of thick low-level clouds and 36 subsequent loss of lidar signal above. You do not know a priori whether the low-level clouds 37 are above or in the BL. Also, consider the miniMPL signal in the free troposphere (above say 38 1.5km) during this day shown in Figure 3b. You only see some backscatteredsignal (light blue 39 colors, well above 0 a.u.) intermittently, indicative of signal return from these altitudes at these 40 times. It's not clear whether the 'hour of day' is UTC or LT, but regardless, the MPL clearly 41 has sufficient power to resolve a background at both daytime and nightime, which makes it 42 very useful.

The miniMPL figure (3b) suggests that all the red colors are likely clouds, given there is no
detectable signal above. This would explain why your gradient method (white circles Figure
3b) fails at these times of red-colored 'clouds', yet seems believable otherwise (e.g.10-12 hours,
15-18 hours – note the cloud at 12-14 hours is very likely sitting at the BL top).

As for the ceilometer results, your gradient method seems to be (successfully) detectingcloud
base height in the present of cloud, or at least, the maximum backscatter signal gradient
inside cloud, but this of course is not BLH. It agrees with the miniMPL gradient algorithm
well during clear air (10-12 hours, 15-18 hours).

It is not clear to me how your IEDA algorithm can work where there is zero signal above 52 optically thick clouds. I can follow how it works when there are no clouds (e.g. Figure 2). In 53 fact, close inspection of Figure 3 (both ceilometer and miniMPL) suggests that the gradient 54 method is working for both instruments (note that both agree during cloud-free periods 55 throughout the day), whereas IEDA varies substantially during cloud-free periods and based 56 on Figure 3, seems the less trustworthy method for either instrument. Thus I disagree with 57 your statements on lines 133 - 135. I suggest that the gradient method is the best (only) one to 58 use for both instruments, based on Figure 3, during cloud-free conditions only.

As a first step to rectifying/addressing these issues, you must positively identify low-levelclouds
and then remove these profiles before you implement either BLH algorithm (especially the 2D
image analyses IEDA) and before any subsequent analyses and BLH statistics are presented.

Although I hope you are in fact doing this, you have not provided clear evidence in the text that
you are indeed removing these cloudy profiles, and I worry that you are still incorporating
cloudy profiles in your results. (see also my Major Comment 2 below)

I suggest in revising Figure 3, you show these plots in logarithmic color scales. This will give the reader (and this reviewer) much more confidence in where you do / do not have lidar and ceilometer signal above what I identify as 'clouds' Also you should identify periods of cloud in these plot yourself too (shade on top perhaps? – there are numerous examples in the literature which you could follow), and confirm in the text that you are removing these cloudy profiles prior to BLH calculation and analyses (at the least, removal of cloudy profiles where optically thick cloud is present in the BL, which preclude any BLH determination).

In summary, as presented in Figure 3 and in the text, these results are likely incorrect and I
strongly urge you to outline and demonstrate an adequate cloud removal algorithm (and
subsequent lidar backscatter profile removal) prior to BLH determination (gradient and IEDA
methods), before Figures 4 and 5 in your manuscript can be trusted. At present, only the
gradient method seems trustworthy and then only for periods where no low-level clouds exist.

**77 **Response:**

Thank you for the suggestion. We have improved the IEDA BLH detection algorithm by
applying cloud removal and revising the process of the image conversion. We hope it can make
the algorithm clearer and more trustworthy.

**81 Cloud removal:**

First, we used the gliding method to identify clouds. According to the characteristics of the cloud backscattered signal (Zuev et al.,1987; Cadet et al., 2005), it can be known that the integral of noise is close to 0, while in the area where the cloud exists, the signal integral value will be a considerable positive one. Therefore, the signal integral in the cloud can be very different from that in other cloudless areas, which can help effectively distinguish the cloud signal from noises. As shown in the following figure, a filtering window W is presented. Wh and WL are the upper and lower edges of the window respectively to integrate the echo signal in the window, i.e

$$C(w) = \int_{W_{L}}^{W} X(z) dz$$
(1)

$$= \int_{W_L}^{W_H} (P(z) + v(z)) z^2 dz = \int_{W_l}^{W_H} P(z) z^2 dz + \int_{W_l}^{W_H} v(z) z^2 dz$$
(2)

$$=\int_{W_L}^{W_H} P(z) z^2 dz$$
(3)

Where C(w) is the integral value of window W, and X (z) is the range-corrected signal. C(w)
value approaches 0 in a certain interval if noises exist. For clouds, the integral value C(w) will
be significantly larger than that of the aerosols. By properly selecting the threshold C(w), the
cloud information can be extracted through the window integral value.

**Figure.** The cloud identification by sliding window integral algorithm window.

Furthermore, in order to ensure that all clouds within the measurement range are extracted, the 98 99 window starts to move from the ground to the maximum detected height. Every time the window 100 moves to a new area, the backscattered signal of the area is integrated to determine whether there 101 is a cloud. If a cloud is present, further calculation of the position of the cloud base and cloud top 102 will be implemented. Next the window continues to move upward from the cloud top until the 103 signal ends. Because the inversion of cloud information is estimated by the movement of the 104 window and the integral calculation of the signal in the window, this algorithm is called sliding 105 window integral algorithm. We then identify the BLH after discarding the detected cloud profiles. 106 Improved IEDA:

Initially the gray conversion was applied to the colormapped image. We found that such108 conversion depends on which channel (red, green, or blue) of the image that the user has decided109 to use. However, none of the channel varies smoothly from black (0) through to white (255).

Therefore, we improved the IEDA algorithm by making the conversion directly from lidar 111 backscatter signal to grey value (0-255), in other words, producing a greyscale pseudocolour plot. 112 The IEDA processing steps are shown in Figure 2. We also described IEDA algorithm more 113 precisely in the section 2.1.1 of the revised paper.

Revised Figure 3

The revised Figure 3 below shows the BLH retrieval from ceilometer and miniMPL against 119 ERA5 reanalysis data under cloudy conditions. For the miniMPL, the IEDA BLH (green line in 120 Figure 3a) accords better with ERA5 BLH (magenta star) and provides fewer variable results 121 compared to the gradient method (black circle in Figure 3a) as expected. For the ceilometer, the 122 gradient method (black circle in Figure 3b), in the contrary, outperforms the IEDA (green line in 123 Figure 3b), with the latter overestimating the BLH significantly compared to the ERA5 BLH. It is 124 possibly because the IEDA fails to distinguish the target area and background area due to the low 125 SNR of the ceilometer. The gradient method, more sensitive to the signal, may be applicable for 126 ceilometer's BLH detection as it operates with a low-powered laser compared to the miniMPL. 127 The discrepancy and uncertainties between the miniMPL/ceilometer and ERA5 can be mainly 128 attributed to (1) the different definitions of the BLHs applied to each method (i.e., edge/gradient 129 detection from aerosol back-scattered signal for miniMPL/ceilometer, bulk Richardson method 130 for ERA5), (2) the different air masses for the spatial separation of the observing sites, (3) the 131 coarse resolution of ERA5 and (4) the presence of the lofted layer or cloud layers. Among these 132 causes, (1) (2) and (3) would influence our whole period. In the case of cloudy condition, we 133 believe the presence of the clouds contributes the most, though other reasons are not to be ruled 134 out.

Therefore, the BLH detection under cloudy conditions remains challenging for the miniMPL 136 and ceilometer even after the cloud removal. We would recommend the BLH retrieval under 137 cloud-free days and specifically IEDA for the miniMPL and gradient method for the ceilometer. 138 This conclusion is not only applicable for cloudy days, but for cloud-free days. More BLH 139 comparison, including the statistical analysis for the whole observing period and specific case 140 study can be found in the section 3.1 (please also see our response to the major comment 3) and 141 section 3.2 in the revised version.